# UDANG: Unsupervised Domain Adaptation with Neural Gating for learning invariant representation of subspaces

## Abstract

The key assumption of deep learning is that the data that the model will be tested on (target domain) are drawn from the same distribution as the data it was trained on (source domain). Breaking this assumption can lead to a significant drop in performance despite having similar underlying features between the source and target domains. Unsupervised Domain Adaptation (UDA) involves using unlabeled samples from the target domain, in addition to labeled samples from source domain, to train a model that can perform well on the target domain. Many existing UDA approaches rely on domain adversarial training (DAT) to reduce domain shift. Although effective, they do not explicitly disentangle the learned features into task-specific and domain-specific components. As a result, the features despite appearing to be domain invariant, may still contain domain-specific biases. To address this, we propose a novel method, UDA with Neural Gating (UDANG), that utilizes a dual adversarial objective to learn an adaptive gating which dynamically route each feature dimension to either the domain or task subspace. Using our strategy, networks have the ability to effectively disentangle task-specific features from domain-specific ones. We validated our approach in multiple datasets and network architectures for image classification, demonstrating strong adaptation performance while retaining the features for discerning the domain.

## 1 Introduction

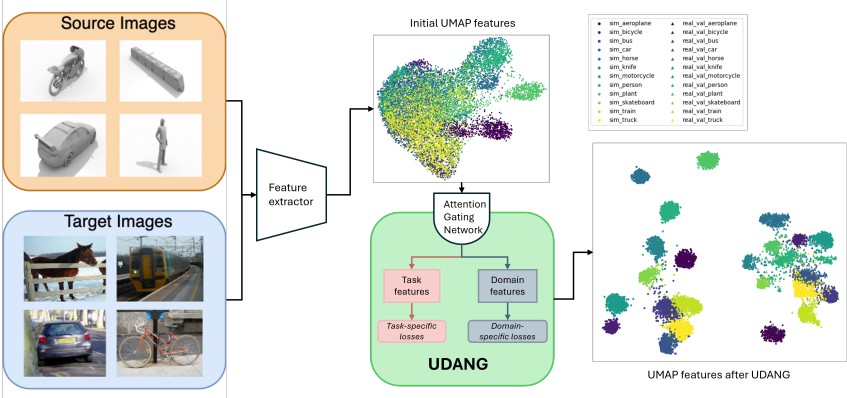

Figure 1: Key idea of the proposed method. We leverage on an attention gating mechanism to dynamically route features learned during training to either task-specific or domain-specific components, which reduces domain-specific biases as shown by the improved separation of classes in both the source (sim) and target (real) domains: (Top) before, (Bottom) after UDANG over the VisDA-2017 dataset. Zoom in for details.

At the core of all modern deep learning is the Empirical Risk Minimization (Vapnik, 1991) objective. In this setting, we assume that data comes from a *fixed* but unknown distributions $q(x)$ and

$q(y|x)$. The goal of learning is then to obtain $\hat{y} = f(x, \theta*)$ whereby some distance $\mathcal{L}(\hat{y}, y)$ is minimized over a dataset of samples drawn from $q$. With a large number of samples, the empirical risk approaches the true risk, giving us a good $f$. While powerful, this framework comes with practical limitations. Firstly, there may be unavoidable differences between training samples, $q_{train}$, and samples encountered by the deployed model, $q_{test}$. For example, in image classification tasks, images captured by a new sensor that was not available when building the training dataset will introduce subtle systematic differences between the samples. A less subtle difference could also arise due to lighting differences, or long term changes in signal properties (distribution drifts). Secondly, this framework is somewhat restrictive as it implicitly requires the train samples to be generated from the same process as the test samples. Solving this problem allows the use of parallel data, e.g. rendered images, where labels can be obtained efficiently for training. In a nutshell, certain properties in a dataset are key to performing well in the desired task, but some are just distractions. While humans perform reasonably well in this setting, deep learning models do not inherently learn to ignore these distracting features. Therefore, undefined or undesirable performance differences *might* happen in the presence of these changes. Our work aims to disentangle the important features pertaining to the task, *task features*, from the distractions, *domain features*.

One common way to deal with distribution shifts is to increase the volume of training data, and this has led to the emergence of foundation models such as DINO (Caron et al., 2021), CLIP (Radford et al., 2021), SAM (Kirillov et al., 2023) that were pre-trained on vast and diverse datasets that spans across multiple domains. While foundation models learn broad representations, they do not inherently address the fundamental problem of domain shift. On the other hand, domain adaptation models actively try to mitigate domain-specific biases by learning domain-invariant representations. Domain adaptation remain necessary for real-world deployment, especially in specialized fields such as medical imaging and robotics where domain shifts are significant and collecting sufficient data to train a foundation model is often impractical.

Learning truly domain-invariant representations is a non-trivial task, particularly when labeled data is unavailable in the target domain. Without labels, there is no explicit supervision to guide feature alignment between the source and target distributions, hence the model must learn to identify which features remain invariant across domains. Advances in UDA, like domain adversarial learning and statistical distribution alignment, have enabled the learning of invariant representations while reducing reliance on domain-specific cues, even in the absence of labeled target data. These techniques have also been applied to related tasks, including multi-source domain adaptation, test-time adaptation.

Many authors rely on domain adversarial training as part of their framework (see Section 2.2). The core idea of domain adversarial training is to generate features that are indistinguishable across source and target domains. This is achieved by having a discriminator that tries to tell the source and target features apart, while feature extractor tries to fool the discriminator. A key limitation is that the feature extractor may only learn to suppress the specific domain biases that the discriminator is sensitive to, leaving other, more subtle domain-specific information within the representations. Disentangling task-specific features from domain-specific features provides a more robust guarantee that the final features utilized for classification are free of domain-specific information.

In this paper, we introduce a novel domain adaptation method that builds upon the DAT framework for disentangled representation learning, see Figure 1. Concretely, we formulate a new objective which employs a dual adversarial strategy to simultaneously learn task-specific and domain-specific information. Moreover, to realize this new objective, we introduce an attention gating mechanism, which dynamically routes each feature dimension to either task or domain subspace. To evaluate the effectiveness of our approach, we performed thorough experiments across various datasets commonly used as benchmarks for the UDA task, including VisDA-2017 and Office-31. Our experiments not only assessed the performance across the datasets but also demonstrated its ability to disentangle feature representations into task-specific and domain-specific features in the latent space.

## 2 RELATED WORK

### 2.1 STATISTICAL ALIGNMENT

Many unsupervised domain adaptation works adopt a statistical approach to align the distributions between source and target domains. Discrepancy metrics such as Maximum Mean Discrepancy (MMD), Wasserstein distance, f-divergence are commonly used to align the marginal distributions (Tzeng et al., 2014; Long et al., 2015; Tzeng et al., 2017; Zhang et al., 2021; Long et al., 2013; Wang & Mao, 2024). The intra-class discrepancy and inter-class discrepancy across domains were further improved in (Kang et al., 2019; Deng et al., 2019; Tang et al., 2020). While these approaches explicitly ensure that statistical properties are consistent across domains, they are often sensitive to variations in sampling from the domains, which can result in mini-batches that do not accurately reflect the overall class distribution. To address this, Kang et al. (2019) employs class-aware sampling, where each mini-batch includes the same subset of classes from both source and target domains, with pseudo-labels for the target data being progressively generated by the network during training.

### 2.2 DOMAIN ADVERSARIAL TRAINING

Instead of explicit alignment using statistical distributions, adversarial-based objectives can be used to implicitly align domain distributions. Domain adversarial minimax game is a technique that involves a two-player game between a feature extractor and a domain discriminator that is optimized using gradient reversal algorithm (Ganin & Lempitsky, 2015; Ganin et al., 2016). The domain discriminator is trained to distinguish whether a given feature representation comes from the source domain or the target domain (minimization) while the feature extractor aims to make the features from both domains indistinguishable to the discriminator (maximization) (Ganin et al., 2016). This technique has been further developed by many through modifications to the optimization framework, *e.g.* task-loss (Rangwani et al., 2022) and minimax objective (Long et al., 2018), and incorporated into their models (Hoyer et al., 2023). One key limitation to domain adversarial learning is that since the feature extractor can create an unlimited number of transformations, it can manipulate target features in a way to make them look like the source features even when they are fundamentally different (Shu et al., 2018). This can be overcome by enforcing the separation of features into task-specific and domain-specific dimensions, as proposed in our method. This forces the model to learn what features contributes to the task and what contributes to domain.

### 2.3 STYLE TRANSFER

Style transfer has been utilized in domain adaptation by transforming the images of the one domain to resemble those from another domain (Hoffman et al., 2018). Certain approaches explicitly construct a new intermediate stylized domain where these methods first transform source images to an intermediate domain with styles from the target domain (Wang et al., 2022; Dundar et al., 2018; Wu et al., 2018). These methods applied statistical style transfer techniques which modify image appearance by aligning color and texture of both domains. Another class of methods is to interpolate between the source and target images to obtain intermediate data samples in order to bridge the gap between the two domains (Xu et al., 2020; Yan et al., 2020). Other methods achieved style transfer effects through consistency loss constraints, several of which (Hoffman et al., 2018; Gong et al., 2019; Li et al., 2019) adopt the CycleGAN (Zhu et al., 2017).

### 2.4 GEOMETRIC SUBSPACE ALIGNMENT

Subspace alignment can be achieved by learning a mapping function that transforms the source subspace into a new one that closely aligns with the target subspace. This is done by minimizing the Bregman matrix divergence between the transformed source subspace and target subspace (Fernando et al., 2014). Works like (Fernando et al., 2013; Mirkes et al., 2022) utilize orthogonality to determine which source basis vectors are relevant for alignment, *i.e.* a source basis vector that is orthogonal to all target basis vectors is disregarded as it is considered to have no shared information with the target domain. (Bousmalis et al., 2016) uses orthogonality differently, where each domain has its own individual subspace, and the shared subspace is constrained to be orthogonal to the individual subspaces.

## 3 PRELIMINARIES

### 3.1 SETUP

Given a source domain dataset $S = \{(x_i^S, y_i^S)\}_{i=1}^{N_S}$ with $N_S$ labeled samples and a target domain dataset $T = \{(x_i^T)\}_{i=1}^{N_T}$ with $N_T$ unlabeled samples, and both domains have the same label space $\mathcal{Y} \in \{1, ..., K\}$ where $K$ is the number of classes. The source data points are sampled i.i.d. from distribution $P_S$, likewise target samples are i.i.d sampled from $P_T$. The objective is to learn a hypothesis $h : \mathcal{X} \rightarrow \mathcal{Y}$ that give accurate target predictions $y_i^{N_T} \in \mathcal{Y}$.

### 3.2 MINIMAX OBJECTIVE

Domain adversarial training (DAT) introduced by (Ganin et al., 2016) offers a practical framework for UDA using a minimax objective and gradient reversal layer, shown in equations 4, 5. For more details on DAT, see Section 2.2. In addition, Acuna et al. (2021) provided a theoretical justification (equations 1, 2, 3) for DAT's effectiveness.

**Theorem 1 (Generalization bound)**. Let the expected source risk be $R_S(h) = \mathbb{E}_{x \sim P_S}[h(x)]$, and expected target risk to be $R_T(h) = \mathbb{E}_{x \sim P_T}[h(x)]$. An upper bound on target risk $R_T(h)$ for hypothesis $h$ is given by:

$$R_T(h) \leq R_S(h) + \delta_{\mathcal{H}}^{\phi}(P_S||P_T) + \epsilon \tag{1}$$

where $\delta_{\mathcal{H}}^{\phi}$ discrepancy is the lower bound estimate of general f-divergences, $\epsilon$ is the irreducible error given by $R_S(h^*) + R_T(h^*)$ with $h^*$ representing the ideal joint hypothesis over both domains.

**Domain Adversarial Objective**. From equation 1, the minimization problem is applied to feature space distributions. Let $h = f_{task} \circ g$, where $g$ is the feature extractor and $f_{task}$ is the object classifier. Also, let $z$ be the output features from $g$. Hence, the objective is:

$$\min_{h \in \mathcal{H}} \mathbb{E}_{x \sim P_S}[L^{task}(f_{task}(z), y)] + \delta_{\mathcal{H}}^{\phi}(P_S^z||P_T^z) \tag{2}$$

where $L^{task}(f_{task}(z), y) = -\sum_{i=1}^{K} y_i \log \hat{y}_i$.

Next, define $D$ as the domain discriminator. Here, $\delta_{\mathcal{H}}^{\phi}(P_S^z||P_T^z)$ can be written as maximizing $L^{dom}$, where:

$$L^{dom}(D \circ g) = -\mathbb{E}_{x \sim P_S}[\log(D(g(x)))] - \\ \mathbb{E}_{x \sim P_T}[\log(1 - D(g(x)))] \tag{3}$$

This leads to the optimization objective given below. The minimax objective forces $g$ to learn a representation where source and target features are as similar as possible, which effectively cause the distance between the two distributions in the latent space to be minimized. This objective is what was used inside DAT.

$$\min_{g, f_{task}} \max_{D} \mathbb{E}_{x \sim P_S}[L^{task}(f_{task} \circ g(x), y)] + L^{dom}(D \circ g) \tag{4}$$

Let $GRL : \mathbb{R}^n \rightarrow \mathbb{R}^n$ denote the gradient reversal layer (GRL). This layer achieves minimax objectives using gradient manipulation. In the forward pass, $GRL$ acts as identity function $GRL(z) = z, \forall z \in \mathbb{R}^n$ while in the backward pass, $GRL$ negates the gradient by a scale factor, i.e. $\frac{dL^{dom}}{dz} = -\alpha \frac{dL^{dom}}{d(GRL(z))}$. Hence, with the GRL layer, minimizing $L^{dom}$ over $D$ can achieve the same max effect:

$$\min_{g, f_{task}, D} \mathbb{E}_{x \sim P_S}[L^{task}(f_{task} \circ g(x), y)] + L^{dom}(D \circ GRL \circ g) \tag{5}$$

## 4 METHOD

In the pursuit of achieving disentangled representation learning, we formulate a new objective to simultaneously learn task-specific and domain-specific information. This leads to two separate groups

of dimensions: domain subspace and task subspace. Domain subspace captures domain-specific information while task subspace captures task-specific information (visualizations are provided in Section 5). In other words, domain subspace comprises of domain-invariant features (*i.e.* task features that are consistent across domains), whereas task subspace constitutes task-invariant features (*i.e.* domain features that are invariant to different object classes). To realize this new objective, we introduce the attention gating mechanism that will dynamically assign each feature dimension to either task or domain subspace. In this section, we will first introduce the new objective function for disentangled representation learning, followed by our model architecture (Figure 2) along with the attention gating mechanism.

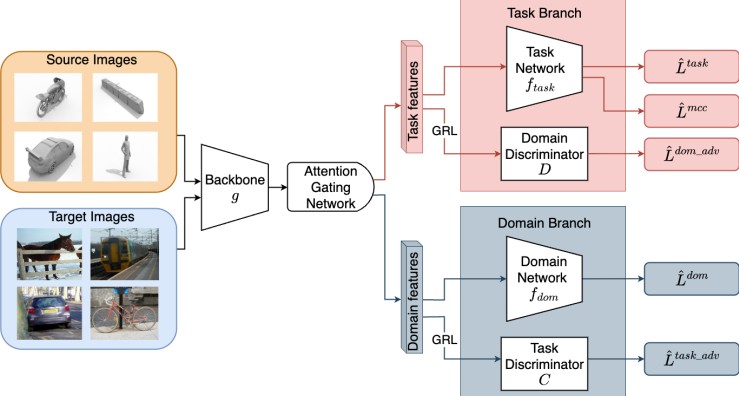

Figure 2: Details of our proposed method. A shared backbone $g$, attention gating network that routes feature dimensions into either the task path or the domain path. The task path contains task network $f_{task}$, and domain discriminator $D$ with $GRL$. Here, the task classifier wants the task features to be discriminative for the task, while the domain discriminator with $GRL$ wants them to be non-discriminative for the domain. This makes the task features invariant to domain. As for the domain path, the domain network $f_{dom}$ tries its best to distinguish between domains, whereas the the task discriminator $C$ together with $GRL$ cause classes to be non-distinguishable, hence this removes any class information from domain features.

### 4.1 DISENTANGLED REPRESENTATION LEARNING OBJECTIVE

The formulation of our novel objective for learning disentangled feature representations extends the domain adversarial minimax objective previously outlined in Section 3.

We denote additional terms: $C$ as the task discriminator, $f_{dom}$ as the domain classifier (not to be confused with domain discriminator $D$). The objective for learning task-invariant features is:

$$\min_{g,f_{dom}} \max_{C} L^{dom}(f_{dom} \circ g) + \mathbb{E}_{x \sim P_S}[L^{task}(C \circ g(x), y)] \tag{6}$$

Moreover, we utilize the minimum class confusion (MCC) loss (Jin et al., 2020) to encourage high-confidence predictions on the target domain samples. Putting equations 4, 6, and MCC together, we have the final minimax objective:

$$\min_{g,f_{task},f_{dom}} \max_{C,D} \mathbb{E}_{x \sim P_S}[L^{task}(f_{task} \circ g(x), y)] + L^{dom}(D \circ g) +$$
$$L^{dom}(f_{dom} \circ g) + \mathbb{E}_{x \sim P_S}[L^{task}(C \circ g(x), y)] + \tag{7}$$
$$\mathbb{E}_{x \sim P_T}[L^{mcc}(f_{task} \circ g(x))]$$

Similar to the domain discriminator, the task discriminator utilizes $GRL$ as well. Hence, the equation is simplified to a min objective:

$$\min_{g,f_{task},f_{dom},C,D} \mathbb{E}_{x \sim P_S}[L^{task}(f_{task} \circ g(x), y)] + L^{dom}(D \circ GRL \circ g) +$$
$$L^{dom}(f_{dom} \circ g) + \mathbb{E}_{x \sim P_S}[L^{task}(C \circ GRL \circ g(x), y)] + \tag{8}$$
$$\mathbb{E}_{x \sim P_T}[L^{mcc}(f_{task} \circ g)]$$

In empirical form with $\hat{L}$ denoting empirical losses:

$$\min_{g, f_{task}, f_{dom}, C, D} \hat{L}^{task}(f_{task} \circ g(x^S), y^S) + \hat{L}^{dom\_adv}(D \circ GRL \circ g(x^S, x^T)) +$$
$$\hat{L}^{dom}(f_{dom} \circ g(x^S, x^T)) + \hat{L}^{task\_adv}(C \circ GRL \circ g(x^S), y^S) + \quad (9)$$
$$\hat{L}^{mcc}(f_{task} \circ g(x^T))$$

In equation 9, $\hat{L}^{task}$ is the empirical task loss on source domain and $\hat{L}^{mcc}$ on the target domain, with both losses applied to task features. $\hat{L}^{dom\_adv}$ is the empirical domain adversarial loss which forces the feature extractor to learn a feature representation where task features in the source and target domains are indistinguishable. These three terms constitute the training losses for the task branch shown in Figure 2. For the training losses in the domain branch, $\hat{L}^{dom}$ serves to distinguish whether features come from the source or target domains. Its opposing force $\hat{L}^{task\_adv}$ discards class information from these domain features. See Appendix A.2 for their full definitions.

### 4.2 GATING MECHANISM

We denote an output feature vector from feature extractor $g$ as $z \in \mathbb{R}^d$. Upon layer normalization with learnable affine transform parameters (Ba et al., 2016) to get $z_{norm}$, it then undergoes linear projection to generate query $q \in \mathbb{R}^d$, key $k \in \mathbb{R}^d$, and value $v \in \mathbb{R}^d$ as shown in equation 10:

$$q = W^q z_{norm}, \ k = W^k z_{norm}, \ v = W^v z_{norm} \quad (10)$$

where $W^q, W^k, W^v \in R^{d \times d}$ are learnable projection matrices.

The Gumbel-Max trick developed for categorical distributions (Gumbel, 1958; Maddison et al., 2014), can be naturally extended to continuous distributions like the Bernoulli. The gates, defined as $m$, are generated from a Gumbel-Sigmoid distribution (see Appendix A.1 for mathematical details). Hard gate is used in the forward pass (equation 11), while a differentiable path is maintained for gradient updates in the backward pass (equation 12):

$$m_{forward} = round(\sigma(\cdot) > 0.5) \quad (11)$$

$$m_{backward} = \sigma\left(\frac{a + \gamma_1 - \gamma_0}{\tau}\right) \quad (12)$$

where $\sigma(x) = \frac{1}{1+e^{-x}}$ is the sigmoid function, $\gamma_0, \gamma_1 \sim Gumbel(0, 1)$, $\tau \in \mathbb{R}^+$ is a temperature parameter and $a = \text{softmax}(\frac{qk^T}{\sqrt{d}})v$, with $d$ as the feature dimension. When $\tau \to 0$, $z$ becomes one-hot.

The task and domain feature dimensions are then obtained by element-wise multiplication of features and their respective gating values, with task gates and domain gates summing to $1 \in \mathbb{R}^d$.

$$z_{task} = m_{forward} \odot z_{norm}, \ z_{dom} = (1 - m_{forward}) \odot z_{norm} \quad (13)$$

After the gating module, there are two distinct branches: task branch and domain branch. $z_{task}$ will go to the task branch while $z_{dom}$ will enter the domain branch. A detailed description is written in Figure 2.

During inference time, the target images will pass through $g$, attention gating network and $f_{task}$. The attention gating network isolates the task-specific features from the domain-specific ones, and only the task-specific features will be used for the classification task.

## 5 EXPERIMENTS

### 5.1 DATASETS & IMPLEMENTATION DETAILS

We used two widely-used UDA image classification datasets for benchmarking—VisDA-2017(Peng et al., 2017) and Office-31(Saenko et al., 2010):

**VisDA-2017**: The VisDA-2017 dataset contains about 280,000 images divided into 12 classes, and has 2 domains: the source and target domain. Its purpose is to facilitate the transition of a model's understanding from simulated to real-world data, particularly in fields like robotics where collecting of real data is often costly and time-consuming.

**Office-31**: The dataset consists of 4,652 images and 31 categories, which are drawn from three domains: Amazon (A), Webcam (W), and DSLR (D). There are six task combinations: A → W, D → W, W → D, A → D, D → A, and W → A which will be evaluated. In real use cases such as inventory management or consumer product recognition, the model trained on clean images (*e.g.* Amazon product catalog) must still be able to accurately identify the same objects in different visual environments, like messy Webcam feeds or high-quality DSLR photos.

We followed the standard protocol used for UDA as in (Rangwani et al., 2022; Long et al., 2018). The mean classification accuracy is calculated based on five independent random experiments. Refer to Appendix B for detailed experimental setup.

## 5.2 EXPERIMENTAL RESULTS

Table 1: Classification Accuracy (%) on VisDA-2017. **Best**, Second best, and *Third best*.

| Method | | Plane | Bcycl | Bus | Car | Horse | Knife | Mcyle | Persn | Plant | Sktb | Train | Truck | Mean |
|---|---|---|---|---|---|---|---|---|---|---|---|---|---|---|
| ResNet (He et al., 2016) | | 55.1 | 53.3 | 61.9 | 59.1 | 80.6 | 17.9 | 79.7 | 31.2 | 81.0 | 26.5 | 73.5 | 8.5 | 52.4 |
| DANN (Ganin et al., 2016) | | 81.9 | 77.7 | *82.8* | 44.3 | 81.2 | 29.5 | 65.1 | 28.6 | 51.9 | 54.6 | 82.8 | 7.8 | 57.4 |
| CDAN (Long et al., 2018) | | 85.2 | 66.9 | 83.0 | 50.8 | 84.2 | 74.9 | 88.1 | 74.5 | 83.4 | 76.0 | 81.9 | 38.0 | 73.9 |
| MCC (Jin et al., 2020) | ResNet-101 | 88.1 | 80.3 | 80.5 | 71.5 | 90.1 | 93.2 | 85.0 | 71.6 | 89.4 | 73.8 | 85.0 | 36.9 | 78.8 |
| SHOT (Liang et al., 2020) | | 94.3 | **88.5** | 80.1 | 57.3 | 93.1 | 94.9 | 80.7 | *80.3* | 91.5 | 89.1 | *86.3* | 58.2 | 82.9 |
| SDAT (Rangwani et al., 2022) | | *95.8* | 85.5 | 76.9 | *69.0* | 93.5 | **97.4** | 88.5 | 78.2 | *93.1* | 91.6 | *86.3* | 55.3 | 84.3 |
| MIC (Hoyer et al., 2023) | | 96.7 | **88.5** | 84.2 | **74.3** | 96.0 | *96.3* | *90.2* | **81.2** | 94.3 | 95.4 | **88.9** | 56.6 | 86.9 |
| CAN (Kang et al., 2019) | | **97.0** | 87.2 | 82.5 | **74.3** | **97.8** | 96.2 | **90.8** | 80.7 | **96.6** | **96.3** | 87.5 | **59.9** | **87.2** |
| UDANG | | 95.7 | *85.7* | 79.6 | 66.4 | *93.9* | 96.5 | **90.8** | 79.9 | *93.1* | 92.9 | 83.6 | *57.4* | *84.6* |
| ViT (Dosovitskiy et al., 2020) | | 97.7 | 48.1 | 86.6 | 61.6 | 78.1 | 63.4 | *94.7* | 10.3 | 87.7 | 47.7 | *94.4* | 35.5 | 67.1 |
| TVT (Yang et al., 2023) | | 92.9 | 85.6 | 77.5 | 60.5 | 93.6 | 98.2 | 89.4 | 76.4 | 93.6 | 92.0 | 91.7 | 55.7 | 83.9 |
| CDTrans (Xu et al., 2022) | ViT-B/16 | 97.1 | *90.5* | 82.4 | 77.5 | 96.6 | 96.1 | 93.6 | 88.6 | *97.9* | 86.9 | 90.3 | 62.8 | 88.4 |
| SDAT (Rangwani et al., 2022) | | *98.4* | 90.9 | 85.4 | 82.1 | *98.5* | *96.3* | 86.1 | *96.2* | 96.7 | 92.9 | 56.8 | | 89.8 |
| MIC (Hoyer et al., 2023) | | **99.0** | **93.3** | *86.5* | **87.6** | **98.9** | **99.0** | **97.2** | **89.8** | **98.9** | **98.9** | **96.5** | **68.0** | **92.8** |
| UDANG | | 98.5 | 90.3 | **87.7** | *79.0* | *97.5* | *98.1* | 94.4 | 84.5 | 95.3 | 97.3 | 94.9 | *62.6* | 90.0 |

Table 2: Classification Accuracy (%) on Office-31. **Best** and Second best.

| Method | | A → W | D → W | W → D | A → D | D → A | W → A | Mean |
|---|---|---|---|---|---|---|---|---|
| ResNet (He et al., 2016) | | 68.4 | 96.7 | 99.3 | 68.9 | 62.5 | 60.7 | 76.1 |
| DANN (Ganin et al., 2016) | | 82.0 | 96.9 | 99.1 | 79.7 | 68.2 | 67.4 | 82.2 |
| MADA (Pei et al., 2018) | | 90.0 | 97.4 | 99.6 | 87.8 | 70.3 | 66.4 | 85.2 |
| CDAN (Long et al., 2018) | ResNet-50 | 94.1 | 98.6 | **100.0** | 92.9 | 71.0 | 69.3 | 87.7 |
| SHOT (Liang et al., 2020) | | 90.1 | 98.4 | 99.9 | 94.0 | 74.7 | 74.3 | 88.6 |
| MCC (Jin et al., 2020) | | **95.5** | 98.6 | **100.0** | 94.4 | 72.9 | 74.9 | 89.4 |
| CAN (Kang et al., 2019) | | 94.5 | **99.1** | 99.8 | **95.0** | **78.0** | 77.0 | 90.6 |
| UDANG | | 94.9 | 99.0 | 99.7 | 94.4 | 77.9 | **77.6** | **90.6** |
| ViT (Dosovitskiy et al., 2020) | | 91.2 | 99.2 | 100.0 | 93.6 | 80.7 | 80.7 | 91.1 |
| CDTrans (Xu et al., 2022) | ViT-B/16 | 96.7 | 99.0 | 100.0 | 97.0 | 81.1 | 81.9 | 92.6 |
| TVT (Yang et al., 2023) | | 96.4 | 99.4 | 100.0 | 96.4 | **84.9** | **86.1** | **93.9** |
| UDANG | | **98.5** | 99.3 | 100.0 | **98.0** | 83.4 | 83.7 | 93.8 |

From the results presented in Tables 1 and 2, UDANG is highly competitive in terms of its performance, comparable to recent SOTA UDA methods such as MIC (Hoyer et al., 2023), SDAT (Rangwani et al., 2022) and TVT (Yang et al., 2023) for the larger ViT-B/16 backbones. For VisDA-2017, UDANG is able to achieve ≥ 90% mean classification accuracy, which is only slightly behind MIC which uses a more elaborate Teacher-Student framework to learn domain-invariant features. For Office-31, UDANG is the SOTA for four of the task combinations: A → W, D → W, A → D, W → D and is highly competitive for the challenging D → A, and W → A tasks. These quantitative results show that the addition of the proposed neural gating method effectively disentangles the task and domain specific representations.

We also show the UMAP visualizations of the features learned before and after application of UDANG in Figure 3. One can see clearly that *both* classes and domains are well separated in the target domain using UDANG.

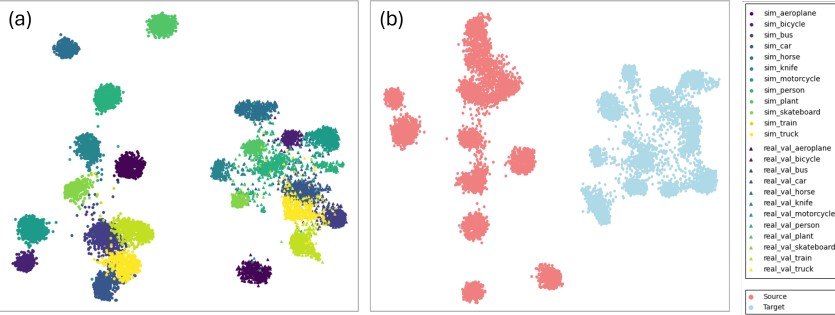

Figure 3: UMAP visualizations of the target domain features using UDANG with ViT backbone on VisDA-2017 dataset showing clear tasks and domain separation. (a) Task separation: Similar classes from different domains are grouped separately. (b) Domain separation: Source and target features are also well separated.

Of interest is the performance of UDANG over specific classes in VisDA-2017 where other methods (e.g. MIC) did not perform well, such as "truck" and "person" which a deeper error analysis is performed, described below.

## 5.3 ERROR ANALYSIS

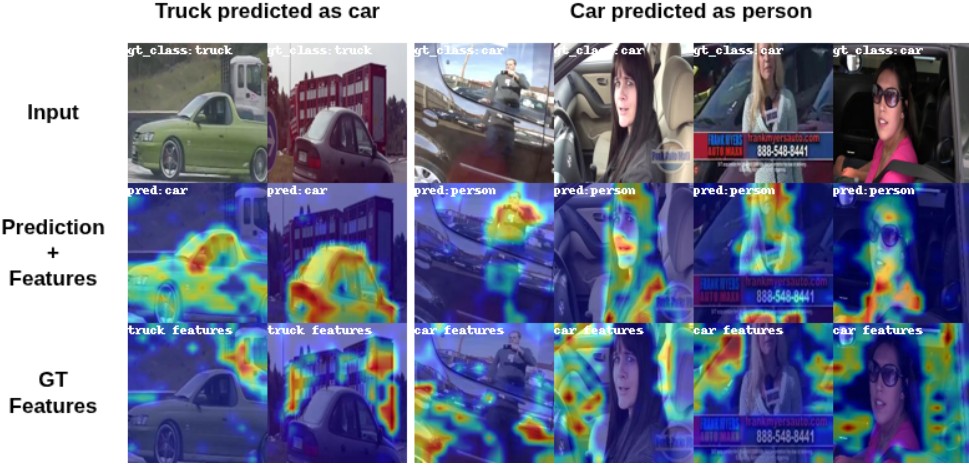

Figure 4: A large number of images in the test set contains two or more relevant classes but are only labeled with one. In the input row, we show samples of misclassified images. The ground truth labels and the predicted labels by our method are overlay-ed on the image. We see that while the predicted labels are different from the ground truth labels, the network is indeed focusing on the relevant regions in the image for the prediction. Indeed, if we were to select the ground truth class for the network to focus on (3rd row), the network focuses on the correct regions in the image for the relavant class.

We report similar findings as MIC where 'mistakes' are often observed in images with multiple subjects. We observe that cars are often predicted as person and trucks are often predicted as cars in the test set. Explainability has been underexplored in the context of UDA, and hence we decided to use GradCAM (Gildenblat & contributors, 2021; Selvaraju et al., 2020) to investigate these failures and determine the source of the class confusion. We noticed that for these classes, the test images for this instances of errors include more than one class of objects in the image. Furthermore, UDANG's

feature selection correctly focuses on the right class when predicting, *i.e.* when predicting a car, it correctly focuses on the features of the car in the image. Additionally, when we studied the features corresponding to the ground truth class, we see that UDANG also correctly selects the features (Figure 4).

## 5.4 OFFICE-31

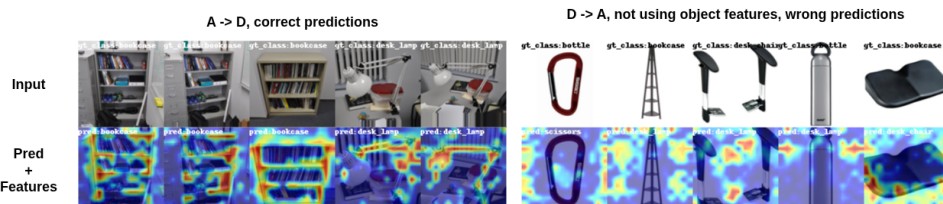

Figure 5: In the left A→D column, we see that the network correctly focuses on the objects, but not the environmental distractions. However, for the D→A adaptation task, plain white background becomes the features that the network focuses on in many of the images. We postulate that this may be due to unintended biases when certain object classes were taken on plain backgrounds.

UDANG obtained SOTA performance on 4 out of 6 permutation of the adaptation task (Table 2), with an overall performance of a close second behind TVT Yang et al. (2023). Interestingly, UDANG demonstrates strong performance when the labeled source dataset is free from distractions from the environment. Upon close analysis with GradCAM, we observe that training with 'clean' images without potentially distracting environmental background enables the network to focus on features on the object. As seen in Figure 5, when training on clean, "Amazon" domain images, the network does not pay attention to environmental clutter when they are prevalent in the test image. However, the converse is not true. When the adaptation is from "DSLR" images to Amazon", the network may be distracted by white background which are not typically seen in many of the "DSLR" domain images. We concede that this is a potential limitation of our method and perhaps semantic cleaning of the source domain image could help adaptation in this direction.

## 6 CONCLUSION

We have presented UDA with Neural Gating (UDANG), that extends standard domain adversarial training (DAT) with an additional dual adversarial objective so that learned features are disentangled into task and domain-specific components. This results in an improved feature representation that is truly invariant across various domains and tasks. Our results and visualizations, in two standard UDA datasets: VisDA-2017 and Office-31, validate the strong performance of UDANG compared to recent SOTA UDA approaches. Additionally, we analyzed the typical failure cases often seen in certain classes, and attribute these failures to the multi-class confusion of the image content itself (e.g. the image contains both "truck" and "person" but is labeled with just one label: "truck").

In future work, we intend to further validate the performance of UDANG across even more domains, even *unseen* ones, a task known as domain generalization and will evaluate the method over larger datasets. Additionally, we will look into combining more elaborate training schemes with stronger semantic guidance to achieve better task and domain disentanglement.

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

# A  MATHEMATICAL DETAILS

## A.1  GUMBEL-SIGMOID

The Gumbel-Max trick Gumbel, 1958; Maddison et al., 2014 is a technique which draws samples $\mathbf{z}$ from categorical distribution with class probabilities $\pi = [\pi_0, \pi_1, ..., \pi_{K-1}]$ where $K$ represents the number of classes and $\gamma_1, ..., \gamma_k$ are i.i.d. samples drawn from $Gumbel(0, 1)$:

$$\mathbf{z} = \text{onehot}(\arg\max_k[\log \pi_k + \gamma_k]) \quad \text{for } k = 0, \dots, K-1 \tag{14}$$

Instead of performing non-differentiable $\arg\max$, softmax is used as an approximation:

$$z_k = \frac{\exp(\frac{\log(\pi_k + \gamma_k)}{\tau})}{\sum\limits_{j}^{K-1} \exp(\frac{\log \pi_j + \gamma_j}{\tau})} \tag{15}$$

In the case of gumbel-sigmoid, we draw samples from Bernoulli distribution with probability $\pi$ of success. The probability mass function is given by:

$$P(z = 1) = \pi, P(z = 0) = 1 = \pi \tag{16}$$

Hence, we have the following where $\gamma_0, \gamma_1$ are i.i.d. samples drawn from $Gumbel(0, 1)$:

$$
\begin{aligned}
z &= \arg\max[\log(\pi) + \gamma_1, \log(1 - \pi) + \gamma_0] \\
&= \frac{\exp(\frac{\log \pi + \gamma_1}{\tau})}{\exp(\frac{\log(1-\pi)+\gamma_0}{\tau}) + \exp(\frac{\log \pi + \gamma_1}{\tau})} \\
&= \frac{\frac{\exp(\frac{\log \pi + \gamma_1}{\tau})}{\exp(\frac{\log(1-\pi)+\gamma_0}{\tau})}}{\frac{\exp(\frac{\log(1-\pi)+\gamma_0}{\tau})}{\exp(\frac{\log(1-\pi)+\gamma_0}{\tau})} + \frac{\exp(\frac{\log \pi + \gamma_1}{\tau})}{\exp(\frac{\log(1-\pi)+\gamma_0}{\tau})}} \\
&= \frac{\exp(\frac{\log \pi + \gamma_1 - (\log(1-\pi)+\gamma_0)}{\tau})}{1 + \exp(\frac{\log \pi + \gamma_1 - (\log(1-\pi)+\gamma_0)}{\tau})} \quad \because \text{Exponent quotient rule:} \frac{\exp(a)}{\exp(b)} = \exp(a - b)
\end{aligned}
\tag{17}
$$

Recall that sigmoid function is $\sigma(x) = \frac{\exp(x)}{1+\exp(x)}$. Let $x = \exp(\frac{\log \pi + \gamma_1 - (\log(1-\pi)+\gamma_0)}{\tau})$. As such, equation is simplified to:

$$z = \sigma(\frac{\log \pi - \log(1 - \pi) + \gamma_1 - \gamma_0}{\tau}) \tag{18}$$

Finally, we express equation 18 in terms of logits. Recall that logit($\pi$)$= \frac{\pi}{1-\pi} = \log \pi - \log(1 - \pi)$, hence we would get:

$$z = \sigma(\frac{\text{logit}(\pi) + \gamma_1 - \gamma_0}{\tau}) \tag{19}$$

## A.2 EMPIRICAL LOSSES

We provide here the full definitions of all empirical losses as defined in 9.

For the task branch:

- $\hat{L}^{task} = \frac{1}{N_S} \sum_{i=1}^{N_S} L_i^{task}(f_{task} \circ g(x_i^S), y_i^S)$

- $\hat{L}^{mcc} = \frac{1}{N_T} \sum_{i=1}^{N_T} L_i^{mcc}$

- $\hat{L}^{dom\_adv} = -\frac{1}{N_S} \sum_{i=1}^{N_S} \log D \circ GRL \circ g(x_i^S) - \frac{1}{N_T} \sum_{i=1}^{N_T} \log(1 - D \circ GRL \circ g(x_i^T))$

For the domain branch:

- $\hat{L}^{dom} = -\frac{1}{N_S} \sum_{i=1}^{N_S} \log f_{dom} \circ g(x_i^S) - \frac{1}{N_T} \sum_{i=1}^{N_T} \log(1 - f_{dom} \circ g(x_i^T))$

- $\hat{L}^{task\_adv} = \frac{1}{N_s} \sum_{i=1}^{N_S} L_i^{task}(C \circ GRL \circ g(x_i^S), y_i^S)$

# B OTHER TRAINING DETAILS

All feature extractors ResNet-50, ResNet-101, and ViT-B/16 were initialized using pretrained weights on ImageNet-1K. For all the experiments, the task network and task discriminator had identical architectures. Also, the domain network and domain discriminator had the same architectures. Across the experiments that used ViT as backbone, 5-layer MLP was used for the task network and task discriminator, and 2-layer MLP for the domain network and domain discriminator. Throughout the experiments with ResNets, we employed a 2-layer MLP for the task network and task discriminator instead. Domain network and discriminator architectures remained the same. Moreover, in all our experiments, we consistently used the same optimizer mini-batch stochastic gradient descent (SGD) and learning rate schedule as Ganin et al. (2016). Gumbel-Sigmoid's $\tau$ was fixed at 1.

**VisDA-2017**: Centre Crop as data augmentation was employed. For VisDA-2017 experiments, ResNet-101 and a ViT-B/16 were used as the feature extractors. In the case of ViT, we trained the model for 30 epochs with a total of 1000 iterations per epoch, and batch size of 32. The initial learning rate is set to 0.0004. The optimizer's momentum and weight decay were 0.9 and 0.001 respectively, which are the same as (Rangwani et al., 2022; Hoyer et al., 2023). We set the MCC loss temperature to be 2, and bottleneck dimension of all the four MLP networks to be 512. In the ResNet-101 scenario, we trained the model for 100 epochs with 2000 iterations per epoch over batch size of 32. We used a temperature of 2 and weight decay of 0.001. The initial learning rate was initialized to 0.001 and bottleneck dimension set to 2048. The optimizer's perturbation radius $\rho$ set to 0.01 in both ViT and ResNet-101 scenarios.

**Office-31**: In Office-31 experiments, ResNet-50 and ViT-B/16 were utilized. The model with ViT backbone was trained for 50 epochs with 100 iterations per epoch over batch size 24. Temperature was assigned to be 1 and momentum to be 0.7. Furthermore, we set $\rho$ at 0.005, and bottleneck dimension at 256. We initialized the learning rate to 0.005. With ResNet-101 as backbone, the model was trained over 250 epochs with 100 iterations per epoch and batch size 24. We initialized the learning rate to 0.009. The momentum and weight decay were 0.6 and 0.001 respectively. Temperature was set to 1, and bottleneck dimension was 512.

