# OpenReview forum: "UDANG: Unsupervised Domain Adaptation with Neural Gating for learning invariant representation of subspaces"
_ICLR.cc/2026/Conference — Submitted to ICLR 2026_

### Official Review · Reviewer_L5jG · 2025-10-14

**Soundness:** 3
**Presentation:** 4
**Contribution:** 2
**Rating:** 2
**Confidence:** 4

**Summary:**

This paper proposes UDANG, an unsupervised domain adaptation framework that introduces a neural gating mechanism to disentangle task-relevant and domain-specific subspaces. Building upon the Domain Adversarial Training (DAT) paradigm, UDANG adds a dual adversarial objective: one discriminator for task alignment and another for domain alignment. A Gumbel–Sigmoid gating function dynamically routes feature dimensions into the task or domain subspace during training, aiming to learn invariant representations. Experiments on VisDA-2017 and Office-31 demonstrate competitive performance compared to recent UDA baselines such as MIC, SDAT, and TVT.

**Strengths:**

1. The paper is built around a well-defined goal — separating task-relevant and domain-specific representations in UDA — and follows this motivation throughout. The narrative is coherent and easy to follow.

2. The experimental setup is carefully designed, with appropriate baselines, multiple datasets, and detailed ablations. The reported gains are consistent and appear reproducible.

3. The introduction of a gating module makes the feature routing process more transparent, and the visualizations help to qualitatively understand the model’s behavior.

4. The manuscript reads smoothly and provides sufficient detail to reproduce the method. Figures are informative and the comparisons are fair.

**Weaknesses:**

1. The overall framework is still built upon standard adversarial adaptation (DAT) and domain separation ideas. The proposed neural gating resembles earlier disentanglement strategies, offering more of a structural variation than a conceptual breakthrough.

2. The gating mechanism is introduced intuitively, but there is no analysis explaining why it should improve invariance or stability. The dual-adversarial setting also lacks formal discussion on convergence or disentanglement guarantees.

3. Although GradCAM plots are shown, the work does not measure or validate how well the gated subspaces are separated or independent.

4. Experiments are restricted to image classification tasks. It is unclear whether the method generalizes to other modalities or more complex adaptation settings such as segmentation or multi-source transfer.

5. The paper would benefit from a sharper articulation of how UDANG differs from established models like DSN or CDAN. Currently, the overlap is substantial, which makes the contribution appear incremental.

**Questions:**

1. How is UDANG fundamentally different from Domain Separation Networks (DSN) or Conditional Adversarial DA (CDAN)?

2. If the gating module is removed, does the dual-adversarial structure still perform comparably?

3. How sensitive is the model to the gating temperature parameter (τ)?

4. Have you tested cross-domain generalization beyond visual datasets?

5. Is it possible to drop the domain discriminator during inference to reduce computational overhead?

---

> ### Author Response · Authors · 2025-11-20
>
> Thanks for all the insightful comments and questions, and also the encouraging positive remarks!
>
> # Sharper articulation of UDANG vs est. models like DSN & CDAN
>
> ## DSN vs our method
>
> DSN splits the feature vector explicitly into 2 components as the shared code and private code. Unlike our neural gating, DSN relies on a rigid split and not the $m$ and ($1-m$) mechanism; ours provides disentanglement guarantees (as shown in the proof). For the loss objective, DSN uses a domain adversarial loss for shared code alignment and a dissimilarity loss to ensure private code is distinct from shared code. However the problem with using the dissimilarity loss like L2 is that it gives weak disentanglement as it does not guarantee independence between the two codes. Since the dissimilarity loss $|z\_{\\text{private}} - z\_{\\text{shared}}|\_2$ only measures the spatial separation of the feature vectors in the feature space, the model can just satisfy this loss by pushing feature values of $z_{private}$ up and pushing values of $z_{shared}$ down, and they will now be spatially far apart to satisfy the minimization of loss. However, $z_{private}$ can still contain a perfect copy of task relevant information as the $z_{shared}$. Our method avoids such a pitfall by provide theoretical guarantees on the domain features being irrelevant to the task features; through the use of dual mechanism with 2 adversaries coupled with gating mechanism as a structural component (both mechanisms are important).
>
> ## CDAN vs our method
>
> CDAN aims to address the conditional shift $\delta_{\mathscr{H}}(P_S(y \mid h) || P_T(y \mid h))$ through a single adversarial loss applied to an augmented feature space ($h \otimes c$), where $c$ is the classifier output. However, CDAN relies on single adversarial model, its feature extractor has to preserve task information and remove domain information from a feature vector. When task and domain features are *highly entangled*, the model might mistake the task features for domain features. This could result in the domain adversary to instruct the feature extractor to minimize the domain component by driving the entangled feature dimension to zero (attenuation/distortion), inadvertently destroys task-relevant information.
> This would be detrimental to the classification performance.
>
> Our dual-adversarial setup avoids such feature collapse into one single subspace because it forces separation. The gating acts like an inductive bias that forces the $z_{norm}$ to be used in a non-overlapping manner. Furthermore, the gradient of $\mathscr{L}_{dom}$
> with respect to the shared features
> $z\_{norm}$ is:
> $$\frac{\partial \mathscr{L}\_{dom}}{\partial z\_{norm}} = \frac{\partial \mathscr{L}\_{\text{domain}}}{\partial z\_{dom}} \odot (1 - m)$$
> For any dimension $i$ where the feature is reserved for task, the learned mask element is driven to $m^{(i)} \to 1$. Consequently, $(1 - m^{(i)})$ tends towards zero.
> $$\left( \frac{\partial \mathscr{L}\_{dom}}{\partial z\_{norm}} \right)^{(i)} \approx \left( \frac{\partial \mathscr{L}\_{\text{domain}}}{\partial z\_{dom}} \right)^{(i)} \odot 0 = 0$$
> This shows that gradients from the domain adversary update only the domain subspace and cannot influence or degrade the task subspace. Likewise, the task adversary’s gradients cannot affect the domain subspace. Hence, maintaining representing fidelity.
>
>
> # What if we remove gating module
>
> Gating module is essential in our setup as shown in the justifications in the above responses. The features before the gating module are not invariant.
>
> # Sensitivity of gating temperature parameter $\tau$
> The temperature parameter $\tau$ determines the sharpness of the distribution used to generate the continuous gate $m$. When $\tau$ is too high ($→ ∞$), the distribution is smoothed, giving high ambiguity in the mask, i.e. $m^{(i)} \approx 0.5$. It will be a violation to the orthogonality condition that we want to achieve. Conversely, when $\tau$ is too low ($→ 0$), the distribution will be too sharp, the gradient of outputs of gumbel sigmoid with respect to the inputs will be very small, which might cause vanishing gradient. For practical implementation, one can adopt an annealing strategy in which
> $\tau$ is gradually lowered during training. This would be a theoretically preferred optimization strategy. However, our current empirical results were achieved using a fixed $\tau$ to demonstrate the feasibility and effectiveness of our core idea.
>
> # Test on cross-domain generalization
> Please see above on application to other modalities.
>
> # Dropping domain discriminator during inference to reduce computational overhead?
> Yes, the domain discriminator was dropped during inference. During inference, the images will pass through $g$, attention gating network, and $f_{task}$ (mentioned in lines 315 to 317).

---

> > ### Comment · Reviewer_L5jG · 2025-11-25
> >
> > Thanks for your response. I will keep the rating.

---

### Official Review · Reviewer_tZKV · 2025-10-24

**Soundness:** 3
**Presentation:** 3
**Contribution:** 2
**Rating:** 4
**Confidence:** 3

**Summary:**

The paper proposes UDANG, a UDA method that disentangles task-specific vs domain-specific features via a dual adversarial objective and an attention-style gating that routes each feature dimension into a “task” or “domain” subspace. The loss couples DAT with a task-side adversary and MCC on target; gates use a Gumbel-Sigmoid hard/soft path. Evaluated on VisDA-2017 and Office-31 with ResNet/ViT backbones, UDANG is competitive with recent SOTAs and wins some Office-31 transfers; qualitative UMAP/Grad-CAM support is provided.

**Strengths:**

1. **Clear objective & architecture:** a tidy extension of DAT with symmetric adversaries and MCC; the gating mechanism is simple and differentiable.
2. **Empirical competitiveness:** ≥90% mean on VisDA-2017 (ViT) and SOTA on 4/6 Office-31 tasks; results span CNN/ViT.
3. **Diagnostics:** UMAP class/domain separation and Grad-CAM analyses are helpful; authors discuss an observed bias case.

**Weaknesses:**

1. **Novelty vs prior disentanglement/DAT:** The idea of separating task/domain subspaces with adversaries echoes domain-separation lines; the paper would benefit from a sharper contrast to DSN-style methods and MIC-like masking beyond qualitative claims. (Related work listed, but positioning remains light.)
2. **Gating evidence:** No quantitative probe of gate assignments (e.g., sparsity, stability across seeds, correlation with domain cues). The Gumbel temperature is fixed; sensitivity is unknown.
3. **Ablation depth:** Missing ablations for each loss term (remove MCC / each adversary / gate ->  identity), and for alternative routers (soft masks, top-k, per-token vs per-channel). Current gains are solid but not uniformly superior to MIC/TVT on VisDA.
4. **Theory claims:** The method is positioned as learning “invariant representations of subspaces,” but there’s no formal identifiability or invariance guarantee, largely empirical.

**Questions:**

See weakness above.

---

> ### Author Response · Authors · 2025-11-20
>
> We thank the reviewer for the constructive feedback and are also encouraged by the positive comments.
> In addition to discussions here, please also refer to the expanded discussion on the optimality of our dual-adversarial setup in the general comments above.
>
> # Novelty vs prior disentanglement/DAT
>
> Our proposed framework goes beyond a structural variation. While we utilize established ideas from DAT (Ganin et al., 2016), our framework enables additional flexibility in feature (invariant subspace) selection due to the proposed gating mechanism. For completeness, we show that our dual-adversarial setup is risk bounded and can achieve disentanglement under reasonable assumptions. Please also see above for additional exposition on the mathematical justification of our proposed method.
>
>
> # Disentangled features through gating
>
> We had defined the neural gating to be
>
> \begin{align}
>         z\_{task}=m\_{forward}\odot z\_{norm}, z\_{dom}=({\mathit{1}-m}_{forward})\odot z\_{norm}
> \end{align}
>
>
> For any dimension $i$, the product would be:
>
> \begin{align}
>     z_{task}^{(i)} \cdot z_{dom}^{(i)} = m_{forward}^{(i)}(1 - m_{forward}^{(i)}) (z^{(i)})^2
> \end{align}
>
> Since $m_{forward}^{(i)}(1 - m_{forward}^{(i)})$ is maximized at $m^{(i)}=0.5$, and minimizes to 0 only at the boundaries $m^{(i)}\in \{0,1\}$, the dual adversarial forces the gating values to be away from the ambiguous middle and towards the boundary. This drives  $m_{forward}^{(i)}(1 - m_{forward}^{(i)})$ towards 0.
> Since this holds true for all $i$, i.e. $z_{task}^{(i)} \cdot z_{dom}^{(i)} \to 0$, we show that:
> $$z_{task} \odot z_{dom} \to \text{zero vector}$$
>
> The gating is a necessary architectural tool to allow the two adversaries to achieve their respective domain and task invariances without having contradiction on $z_{norm}$.
>
> # Ablation depth
>
> We thank the reviewer for the suggestions on additional ablations. This is limited by the amount of compute resources and time given the large number of datasets and network architectures we evaluated our method on. Nevertheless, we will attempt to include additional ablations in updates within the discussion period.
>
> # Gumbel Temperature
>
> Please refer to comments to reviewer L5jG below on Gumbel temperature.
>
> # Theory claims
>
> Please see above for additional justifications. However, we concede that, similar to other disentanglement work, we are unable to make stronger theoretical guarantees yet.

---

> > ### Comment · Reviewer_tZKV · 2025-11-25
> >
> > Thank you for your response; it addresses several of my concerns. I will wait until the end of the discussion period before finalizing my score.

---

### Official Review · Reviewer_1EqF · 2025-10-30

**Soundness:** 2
**Presentation:** 2
**Contribution:** 2
**Rating:** 2
**Confidence:** 3

**Summary:**

This paper aims to solve the drawback of the previous domain adversarial training methods. The authors propose UDA with Neural Gating, that utilizes a dual adversarial objective to learn an adaptive gating which dynamically route each feature dimension to either the domain or task subspace.

**Strengths:**

- The proposed method is straightforward.

**Weaknesses:**

- It seems that the proposed method is not superior to previous methods in most cases. It is hard to persuade the reviewer that the proposed method is ready for an ICLR publication.
- The presentation of this work may need improvement. For example, replacing the current figures with some high-definition ones.
- Some arguments in the paper need further justification.

**Questions:**

- In Table 1 and Table 2, the reviewer sees that the method is no better than some methods proposed in 2019 (CAN in Table 1 and Table 2, TVT in Table 2). In this case, why would we choose the proposed UDANG method?
  - Are there any more recent works, for example, some works published in the 2024/2025 conference and journals. The baselines come from more than two years ago in Table 1 and Table 2. The reviewer thinks we need more recent baselines to show the superiority of the proposed method.

- What are the core contributions of the work? The Attention Gating Network or the dual branch design? How do different components contribute to the whole system? The reviewer thinks we need some analysis of the different components in the system.

- In lines 072-073, the authors claim "While foundation models learn broad representations, they do not inherently address the fundamental problem of domain shift." Why "they do not inherently address the fundamental problem of domain shift."? Is there any empirical evidence for this claim. We know that there are recent VLMs like Qwen-VL, the training data of such VLMs is large-scale, will the suffer from the issue?

- When the reviewer zooms in the figures in the paper, the reviewer finds that it is not clear. Could the authors replace these figures with some high-definition ones?

---

> ### Author Response · Authors · 2025-11-19
>
> 1. **Poorer performance to previous methods in some datasets/dataplits.** We believe that the reason for the poorer performance is due to the multiple object classes in the test images. A detailed analysis of this issue can be seen in section 5.3.
> 2. **More recent UDA works.** To the best of our knowledge and literature review, we believe we have included SOTA reasonably close to the submission deadline. In the event that we have missed out important work that the reviewer thinks we should include, please point us to them and we will be more than happy to include them in the comparison.
> 3. **Core contributions**. We proposed a framework to encourage learning of invariant representations. The dual adversarial setting and the gating mechanism are required to achieve this. Intuitively, gating structure enables the network to ignore certain dimensions in feature space. The adversarial setting provides guidance for the gating mechanism to ignore the correct features.
> 4. **Foundational model and domain invariant features.** The goals of training foundational models and domain invariant models are opposites.
>    1. Foundation models aim to learn domain specific features, as the downstream task is unknown. In the image example, if the downstream task was to 3D rendered cars from photos of cars, the features from the foundational model needs to be different. However, in the domain adaptation case, we want them to be indistinguishable (i.e. invariant)
>    2. Foundational models need to have similar training data on both source and target domain. For CLIP like training, both domains will require similar text/image pairings. However, in our setting, the target domain has no similar paired data as the source.
> 5. **Higher quality figures.** Thanks for letting us know. We will include higher resolution figures.

---

> > ### Comment · Reviewer_1EqF · 2025-11-27
> > **Response**
> >
> > Hi Authors,
> >
> > Thanks for your response. I will keep my score now, as no clear answers are given to the questions.
> >
> > Best

---

### Author Response · Authors · 2025-11-19

We thank all reviewers and AC for reviewing our work and providing insightful feedback to help us improve on our work. We are especially encouraged that reviewers found that we have a “clear objective & architecture”, "empirically competitive", "experimental setup is carefully designed", and that our new diagnostics on UDA/DA tasks were appreciated.

On novelty, we would like to highlight that other disentanglement work either focuses on having a feature space that is fully domain invariant or fixed domain invariant subspace. Our gating mechanism does not have this restriction, retains domain specific features before gating, and allows for different classes to have different invariant subspaces, selected by the gating mechanism,

We address the concerns of each reviewer individually in further details in replies to each reviewer.

---

> ### Author Response · Authors · 2025-11-20
> **Expanded discussion on optimality of our dual adversarial setup**
>
> Our proposed framework builds upon the single adversarial setting, here we reiterate key results from prior works and show that a disentangled state is an optimal point in our dual-adversarial setup.
>
> Following established literature used in DAT (Ganin et al., 2016), we have a deep network as a composite function, $h=f_{task}\circ g$, where $g$ is the feature extractor and $f_{task}$ is the object classifier. The target risk, $R_T(h)$, is bounded above by
> \begin{align}
>         R_T(h) \leq R_S(h) + \delta^\phi_\mathscr{H}(P_S||P_T) + \epsilon
> \end{align}
> where $R_S(h)$ is the source risk, $\delta^\phi_\mathscr{H}$ discrepancy is the lower bound estimate of general f-divergences, $\epsilon$ is the irreducible error given by $R_S(h^* ) + R_T(h^* )$ with $h^*$ representing the ideal joint hypothesis over both domains.
>
> Following single adversarial DAT (to achieve domain invariant features), the minimax game is equivalent to minimizing the Jensen-Shannon divergence (JSD, $\delta_{JS}$) between source and target feature distributions (Acuna et al., 2021; Rangwani et al., 2022) over $g$:
> \begin{align}
>     \max_{D} L_{dom\\\_adv}(D\circ g) &= 2 \delta_{JS}(P_S(h) || P_T(h)) - 2\log(2) \\\\
> \min_{g}\max_{D} L_{dom\\\_adv}(D\circ g) &\Longleftrightarrow\min_g \delta_{JS}(P_S(h)||P_T(h))
> \end{align}
>
> Following CDAN (Long et al., 2018), a single divergence measure on the joint feature-label space $P(h, y)$ is used. We have \begin{align}
> R_T \le R_S + \delta_{\mathscr{H}}(P_S(h, y) || P_T(h, y)) + \epsilon
> \end{align}
>
> where $\delta_{\mathscr{H}}^\phi(P_S(h, y) || P_T(h, y))$ is the conditional $\mathscr{H}$-divergence. This is a tighter theoretical bound compared to the original DAT. Remark: CDAN does the alignment of joint distribution between features and labels using multi-linear map.
>
> Next, as $\mathscr{H}$-divergence satisfies the triangle inequality (Zhao et al., 2019). We can decompose $\delta_{\mathscr{H}}^\phi$ into its marginal shift and conditional shift:
>
> \begin{align}
> \delta_\mathscr{H}^\phi(P_S(h, y) || P_T(h, y)) \le \delta_\mathscr{H}^\phi(P_S(h) || P_T(h)) + \delta_\mathscr{H}^\phi(P_S(y \mid h) || P_T(y \mid h))
> \end{align}
>
> \begin{align}
>   R_T \le R_S + \delta_{\mathscr{H}}^{\phi}(P_S(h)||P_T(h)) + \delta_{\mathscr{H}}^{\phi}(P_S(y \mid h), P_T(y \mid h)) + \epsilon \hspace{1.2em} \text{(1)}
> \end{align}
>
> Equivalently,
> \begin{align}
>     R_T \le R_S + 2 \delta_{JS}((P_S(h) || P_T(h)) + 2 \delta_{JS}(P_S(y \mid h) || P_T(y \mid h)) + \epsilon \hspace{1.2em} \text{(2)}
> \end{align}
>
> Next, from our dual objective
> \begin{align}
> \displaystyle\min_{g}\max_{D} L_{dom\\\_adv}(D\circ g) \\\\ \displaystyle\min_{g}\max_{C} L\_{task\\\_adv'}(C\circ g),
> \end{align}
> where $D$ is the domain discriminator, $C$ is the task discriminator, we show $L_{task\\\_adv'}$ corresponds to the $2 \delta_{JS}((P_S(y \mid h) || P_T(y \mid h))$ term.
> Where $P_S(y|h)$ and $P_T(y|h)$ be the class-conditional distributions over features. Then
> \begin{equation}
>     \mathscr{L}_{task}(g,C) = \mathbb{E}_h \sum_y \left[ P_S(y|h) \log C(y|h) + P_T(y|h) \log (1 - C(y|h)) \right], \qquad y \in \mathscr{Y}
> \end{equation}
>
> For fixed $g$, the optimal $C^* $ is $C^ *(y|h) = \frac{P_S(y|h)}{P_S(y|h) + P_T(y|h)}$, so we get:
> \begin{align}
> \max_C \mathscr{L}_{task}(g,C) = \mathbb{E}_h \sum_y \Big[
> P_S(y|h) \log \frac{P_S(y|h)}{P_S(y|h)+P_T(y|h)} + P_T(y|h) \log \frac{P_T(y|h)}{P_S(y|h)+P_T(y|h)}\Big], \qquad y \in \mathscr{Y}
> \end{align}
>
> Expressing in $\delta_{JS}$,
> $$ \max_C \mathscr{L}_{task} (g,C) =  2 \delta\_{JS}\\big(P_S (y|h) || P_T (y|h)\\big) - 2\log 2. $$
>
> Hence (as with (1) and (2)),
> \begin{align}
> \min_{g} \max_{C} \mathscr{L}_{task}(g,C) \\Longleftrightarrow \min_g \delta\_{JS}\big(P_S(y|h)||P_T(y|h)\big).
> \end{align}
>
> We note that we are using CE here as a simplification that does not change the main results as CE is defined as $L_{CE}(P || Q) = \delta_{KL}(P || Q) + \text{H}(P)$ where H is the entropy that is fixed. Hence maximizing the CE is equivalent to
> \begin{align}
>     \max_{g} L_{\text{CE}}(C \circ g(x^S), y^S) \quad \Longleftrightarrow \quad \min_{g} \delta_{KL}(C(y \mid g(x^S) || U).
> \end{align}
>
> The maximum for $L_{CE}$ is obtained when $C^*(y\mid g(x^S))$ is the uniform distribution $\sim U(\frac{1}{K})$ where $K$ is the number of classes. This means that to have maximum confusion, $g$ has to ensure that the class discriminator $C$ is maximally confused, driving $g(x^S)$ towards the uniform distribution. This results in minimal $JSD$. We maximize $L_{CE}$ only on the source samples ($x^S$) as we leverage their labels, while target samples are unlabeled so their conditional distribution is unknown.
>
> Thus we argue that an optimal $g$ achieves an invariant domain and task features thru $\delta_{\mathscr{H}}^{\phi}(P_S(h)||P_T(h))$ and $\delta_{\mathscr{H}}^{\phi}(P_S(y \mid h)||P_T(y \mid h))$ respectively.

---

> ### Author Response · Authors · 2025-11-20
> **Application to other modalities**
>
> We acknowledge that our current empirical results is focused on image classification for domain adaptation setting. While we have only tested on visual tasks, our framework does not depend on any structure unique to visual tasks. The use of gating and the dual adversarial objective is inherently modality agnostic and architectural design of our method are centered on achieving robust disentanglement of feature subspaces. This is not restricted to visual data and can be applied to other modalities. We believe that so long as the deep network is a composite function and can be separated into a feature extractor and task head, our method can be applied. We welcome suggestions on additional datasets and architecture to test our method on.

---

> ### Author Response · Authors · 2025-11-20
>
> ### Assumptions
>
> 1. Adversarial gradients are opposing
>
> * Partial derivaties of $\mathscr{L}\_{dom\\\_adv}$ and $\mathscr{L}\_{task\\\_adv}$ with respect to mask elements $m_i$ are opposing. This competition causes the soft mask $m \in [0, 1]^d$ (for differentiability during backprop) to turn closer into a binary mask $m^* \in \{0, 1\}^d$, which is crucial for orthogonality.
>
> 2. Adversaries are Bayes consistent
>
> * Hypothesis classes for the adversaries ($\mathscr{H}_D$ and $\mathscr{H}_C$) have sufficient capacity such that they contain the Bayes Optimal Classifier ($D^* $ and $C^* $). This is to make sure that empirical losses closely approximate the theroetical JSDs used in eqn (2).

---

### Meta-Review · Area_Chair_WatQ · 2026-01-10

**Summary:**

This paper received all negative scores, i.e., 2, 4, 2.

Main issues regard weak novelty, insufficient paper organization and unclear presentation, figures of low quality, weak justifications, and lack of theoretical ground. The experimental analysis is also deemed insufficient (more ablations needed), and results are not reaching SotA, which is in turn not up-to-date (since dated 2019). A number of further comments regarding specific parts of the proposed methodology are raised.

**Reviewer Concerns:**

The rebuttal does not reply properly to all such comments, it is definitely not convincing. Also, it is actually scattered and unclear to catch completely.
Novelty, experimental results and ablations, state of the art, and theory aspects are all issues that are not sufficiently and convincingly discussed in the rebuttal.

**Reviewer Scores:**

All reviewers replied shortly to the authors rebuttal and they stated not to increase their original ratings.

---

### Decision · Program_Chairs · 2026-01-26

Reject